# The Involvement of Ubiquitination and SUMOylation in Retroviruses Infection and Latency

**DOI:** 10.3390/v15040985

**Published:** 2023-04-17

**Authors:** Taizhen Liang, Guojie Li, Yunfei Lu, Meilin Hu, Xiancai Ma

**Affiliations:** 1State Key Laboratory of Respiratory Disease, Guangzhou Medical University, Guangzhou 511400, China; 2Guangzhou Laboratory, Guangzhou International Bio-Island, Guangzhou 510005, China; 3Zhongshan School of Medicine, Sun Yat-sen University, Guangzhou 510080, China

**Keywords:** ubiquitination, SUMOylation, retrovirus, viral infection, latent infection

## Abstract

Retroviruses, especially the pathogenic human immunodeficiency virus type 1 (HIV-1), have severely threatened human health for decades. Retroviruses can form stable latent reservoirs via retroviral DNA integration into the host genome, and then be temporarily transcriptional silencing in infected cells, which makes retroviral infection incurable. Although many cellular restriction factors interfere with various steps of the life cycle of retroviruses and the formation of viral latency, viruses can utilize viral proteins or hijack cellular factors to evade intracellular immunity. Many post-translational modifications play key roles in the cross-talking between the cellular and viral proteins, which has greatly determined the fate of retroviral infection. Here, we reviewed recent advances in the regulation of ubiquitination and SUMOylation in the infection and latency of retroviruses, focusing on both host defense- and virus counterattack-related ubiquitination and SUMOylation system. We also summarized the development of ubiquitination- and SUMOylation-targeted anti-retroviral drugs and discussed their therapeutic potential. Manipulating ubiquitination or SUMOylation pathways by targeted drugs could be a promising strategy to achieve a “sterilizing cure” or “functional cure” of retroviral infection.

## 1. Introduction

The *Retroviridae*, which is also called the retrovirus family, can cause many acute or chronic diseases within infected vertebrates. Several “acute transforming retroviruses”, such as the Rous sarcoma virus (RSV), induce rapid tumor formation [1]. Most “slow retroviruses”, such as the murine leukemia virus (MuLV), induce tumorigenesis within chronic latent infection [2]. Apart from mutagenesis, some retroviruses, such as lentiviruses, show significant cytopathic pathogenicity. One of the most notorious cytopathic retroviruses is the human immunodeficiency virus type 1 (HIV-1), which causes acquired immunodeficiency syndrome (AIDS) [3,4]. Although more than 40 years have passed since the discovery of HIV-1, we have not yet found proper strategies to thoroughly eradicate viruses from infected individuals. The biggest obstacle to purging retroviruses from the host is that most retroviruses form stable latent reservoirs within the host [5]. During the viral latency, few viral proteins are expressed, which enables viruses to compromise the effectiveness of anti-retroviral drugs or evade immune surveillance.

Retroviruses were originally classified into four types, including A-type, B-type, C-type, and D-type viruses based on the morphology of their virion cores visualized under the electron microscope. Recently, the retroviral genera have been formalized by the International Committee on Taxonomy of Viruses, which includes “simple” retroviruses and “complex” retroviruses. The simple viruses, including alpharetroviruses, betaretroviruses, and gammaretroviruses, encode only the Gag, Pro, Pol, and Env proteins. Additionally, the complex viruses, including deltaretroviruses, epsilonretroviruses, lentiviruses, and spumaviruses, encode additional regulatory proteins besides the above gene products [6]. These viral particles contain genomic RNAs that are reverse-transcribed into double-strand DNAs upon viral entry. Then, reverse-transcribed DNAs are further integrated into the host genome and present as proviruses, which express viral genomic RNAs and proteins. The reverse-transcription property is the typical feature of retroviral life cycles.

Different retroviruses have unique life cycles utilizing different mechanisms. The reverse-transcription procedure can happen prior to budding or capsid release for different retroviruses. Different retroviruses also utilize a variety of mechanisms for RNA export. Since the life cycle of HIV-1 has been extensively investigated, we next introduce the life cycle of HIV-1 for illustration. In general, the HIV-1 life cycle can be summarized into 10 steps [7]. Firstly, the infectious viral particle binds to the cellular receptor utilizing the envelope protein (Step 1), followed by the membrane fusion of viral membrane and cellular membrane (Step 2) [8]. Upon entering the cytoplasm, the capsid core is gradually disassembled, which is also called the viral uncoating procedure (Step 3) [9]. The exposed viral RNA is reverse-transcribed to form the double-strand DNA (Step 4), which is further transported into the nucleus utilizing the pre-integration complex (PIC) (Step 5) [10]. Then, the double-strand DNA integrates into the host genome and forms the provirus (Step 6) [11]. Both spliced and unspliced viral RNAs are transcribed from the provirus (Step 7), which are exported from the nucleus to the cytoplasm through Rev-responsive element (RRE)-independent and dependent pathways (Step 8) [12]. Within the cytoplasm, these RNAs are translated into both structural and non-structural proteins, or served as the viral genomic RNAs (Step 9) [13]. Finally, these viral proteins and the genomic RNAs are assembled to pack into the particle, followed by the budding, release, and maturation of the virions (Step 10) [14]. Apart from the above active life cycle, retroviruses acquire an inactive life cycle, which is also called latent infection [15]. Upon integrating into the host genome, proviruses temporarily silence within infected cells and produce no viral particles. Once the infected cells are activated by the external stimulus, these latent proviruses can return to the active life cycle and produce large amounts of infectious virions. Both viral and cellular factors contribute to each step of life cycle and determine the fate of active or latent life cycles [16]. Thus, many individual or systematic researches have been conducted to fully elucidate the mechanisms of retroviral infection or latency [17,18].

The infection and latency of retroviruses involve multiple stages extensively modulated by the virus–host interplay. Retroviruses can utilize viral or cellular factors to facilitate their replication and latency. Meanwhile, the host cells can also manipulate intracellular anti-retroviral systems to confront viral infection. Both post-transcriptional regulations of RNAs and post-translational modifications (PTMs) of proteins play key roles in each step of the retroviral life cycle [19,20]. Numerous host-encoded microRNAs and long non-coding RNAs (lncRNAs) have been found to enhance or suppress retrovirus infection and latency. For example, miR-28, miR-125b, miR-150, miR-223, and miR-382 target the 3’ ends of HIV-1 mRNA to promote HIV-1 latency [21]. The lncRNA noncoding repressor of NFAT (NRON) orchestrates the proteasome system to degrade HIV-1 Tat, which also results in quickly entering into the viral latency [22]. Apart from those host-derived non-coding RNAs (ncRNAs), retroviruses also encode various ncRNAs to safeguard their replication and pathogenicity within host cells. The bovine leukemia virus (BLV) encodes the miRNA BLV-miR-B4 to promote BLV-induced B-cell tumorigenesis [23]. In addition, an HIV-1-derived antisense lncRNA promotes HIV-1 latency by ensuring the suppressive epigenetic modifications on the viral promoter [24].

PTMs of proteins shape a more complicated reciprocal network upon the cross-talking of viral and cellular proteins. More than 430 different kinds of PTMs have been identified [25], and the most widely studied PTMs are phosphorylation, ubiquitination, methylation, acetylation, glycosylation, SUMOylation, palmitoylation, myristoylation, prenylation, and sulfation [26]. Almost all the major PTMs have been found to participate in the regulation of retrovirus infection or latency [27]. The phosphorylation of avian retrovirus nucleocapsid protein pp12 enhances the binding affinity of pp12 to viral RNA [28]. Both DNA methylation and histone methylation on the HIV-1 LTR contribute to HIV-1 latency [29,30]. The acetylation of histone or HIV-1 Tat has been found to promote HIV-1 transcription and reactivation [31,32,33]. The high levels of N-linked glycosylation of HIV-1 envelope proteins protect viruses from immune surveillance [34]. In addition, the palmitoylation modifications of both the feline immunodeficiency virus (FIV) and HIV-1 envelope proteins have been found to facilitate the incorporation of envelope proteins into the virions [35,36]. Several studies also demonstrate that the myristoylation of gag proteins of HIV-1 and the human T-cell lymphotropic virus type 1 (HTLV-1) facilitate viral replication and assembly [37,38,39]. Treatment with the prenylation inhibitor GGTI-298 potentially inhibits HTLV-1 LTR transcriptional activity and decreases Tax protein expression, although the mechanism of prenylation to regulate HTLV-1 infection has not been identified [40]. The tyrosine sulfation at the N-terminal of CCR5 or CXCR4, which is catalyzed by the tyrosylprotein sulfotransferase (TPST), has been found to promote the HIV-1 entry [41]. Moreover, the tyrosine sulfation in the second variable loop (V2) of the HIV-1 envelope also has been shown to enhance V2-V3 stability and modulate divergent sensitivities to different neutralization antibodies [42].

Apart from the above eight major PTMs, another two vital PTMs, ubiquitination and SUMOylation, shape a more complicated viral regulation landscape. In this review, we comprehensively summarize past and recent advances in ubiquitination- and SUMOylation-mediated retroviruses infection and latency. The ubiquitination and SUMOylation of both cellular and viral proteins in regulating retroviral life cycle are discussed. In the end, we also discuss the therapeutic potential of targeting ubiquitination and SUMOylation pathways to treat retroviral infection, especially HIV-1 infection and latency.

## 2. Ubiquitination and SUMOylation Pathways

The ubiquitination of a specific protein is a process of the ubiquitin (Ub) molecules (or proteins) covalently conjugated to lysine (Lys) residues on target proteins, which mediates target protein degradation or modulates the protein substrate into a functional signaling molecule. The ubiquitination pathway plays roles in regulating different aspects of cellular or viral activities, including protein degradation, DNA repair, the ER-associated degradation pathway, receptor endocytosis, apoptosis, and autophagy [43,44,45,46,47,48,49]. The conjugation of Ub to target proteins is a multi-step reaction involving three crucial enzymes: the ubiquitin-activating enzyme, E1; the ubiquitin-binding enzyme, E2; and the ubiquitin ligase, E3 [50]. The Ub is a small regulatory molecule (8.5 kDa) encoded by *UBB*, *UBC*, *UBA52*, or *RPS27A* genes within humans [51]. In vertebrates, two enzymes of E1 participate in the activation of Ub, which are UBA1 and UBA6. The E2 enzymes contain nearly 40 different types, about 35 of which are the determinants of the conjugation of Ub to substrates. There are over 600 E3 ubiquitin ligases, which can be categorized into three families: the Homologous to E6-associated Protein C-terminus (HECT), the Really Interesting New Gene (RING), and the U-box Domain [51,52,53].

The ubiquitination process begins with the ATP-dependent activation of C-terminal glycine residue of Ub by a specific activating enzyme E1 (Figure 1). In this step, the Ub molecules bind to the Cysteine (Cys) residue of E1, which requires the ATP hydrolysis to supply the energy. The activated Ub is then transferred to the active site of the E2 enzyme, resulting in the conjugation of Ub on the Cys residue of E2. Subsequently, the E3 ubiquitin ligase interacts with both the Ub-loaded E2 and the substrate protein, which results in the isopeptide linkage between the C terminus of Ub and the specific Lys residue of the substrate [50]. The Lys residues of the substrate protein can be conjugated with a single Ub (mono-ubiquitination) or several Ub molecules (multiple mono-ubiquitinations). Additionally, different Ub chains can be formed by linking several Ub moieties through one of the seven lysine residues (Lys6, Lys11, Lys27, Lys29, Lys33, Lys48 and Lys63) or the N-terminal methionine residue (M1) within the Ub molecule [54,55]. The substrate-conjugated Ub molecules can also be modified by other PTMs, such as phosphorylation, acetylation, and SUMOylation [56,57]. The successful ubiquitination process relies on the strict control of the activities of deubiquitylating enzymes (DUBs). DUBs belong to a diversified family of isopeptidases that make the Ub reusable by removing conjugated Ub tags from substrates [58,59]. There are approximately 20 DUBs in *Saccharomyces cerevisiae* and 100 DUBs in humans [59,60,61]. Currently, DUBs can be subdivided into six subfamilies: the ubiquitin-specific proteases (USPs), the ovarian tumor proteases (OTUs), the ubiquitin C-terminal hydrolases (UCHs), the Machado–Josephin disease protein (MJD), the motif interacting with ubiquitin-containing novel DUB family (MINDY), and the JAB1/MPN/MOV34 metalloprotease (JAMMs, also known as MPN+) [59,60,62]. The USP, OTU, UCH, MJD, and MINDY are Cys proteases, which need two or three crucial amino acid residues to form catalytic dyad or triad. In addition, the deubiquitylation function of the JAMM/MPN+ family relies on the coordination of zinc ions within the active sites [63]. The USP, UCH, and JAMMs can act on both Ub and ubiquitin-like (Ubl) modifications, while other DUBs are specific for Ub only (Figure 1) [64].

The SUMOylation modification is another vital PTM, which is also called the Ubl modification. Its mechanism is highly similar to the ubiquitination procedure [65]. Within most situations, SUMOylation does not induce the degradation of target proteins, but serves as the specific signal for protein-protein interactions. The modifier molecules in SUMOylation are small ubiquitin-like modifiers (SUMOs). Five SUMO molecules (SUMO1, SUMO2, SUMO3, SUMO4, and SUMO5) among vertebrates have been discovered [66,67,68,69,70]. SUMO2 and SUMO3 are typically regarded as the same molecule named as SUMO2/3 in most studies because of the high similarities in their sequences and functions [71]. Similar to ubiquitination, the maturation and the conjugation of SUMO molecules also require multiple enzymes [72]. The SUMO protein is expressed as an immature precursor molecule, which needs to be hydrolyzed and processed into a mature form by SUMO-specific proteases (SENPs) to expose diglycine residues. The C-terminal diglycine motif is the hallmark of mature SUMO. (Figure 2). The processed mature form of SUMO is activated by the E1 enzyme complex (SAE1/UBA2), followed by transferring to the E2 enzyme UBC9. The SUMO molecule on E2 can be modified to the target protein directly, or transferred to an E3 enzyme, which enables or facilitates the conjugation of SUMO molecules to target proteins. These SUMO molecules on SUMOylated proteins can interact with other proteins by specifically binding to the SUMO-interacting motifs (SIMs) presented on interacting proteins [73]. SENPs also deSUMOylate protein substrates to reverse SUMO conjugation and play critical roles in maintaining the balance of SUMOylation/deSUMOylation level of target proteins. To date, seven human SENPs have been identified (SENP1, SENP2, SENP3, SENP5, SENP6, SENP7, and SENP8). In addition to the maturation of SUMO molecules, SENPs can also deconjugate SUMO molecules from target proteins and recycle the removed SUMO molecules (Figure 2) [74]. Although the proportion of SUMOylated proteins is only less than 5% for most protein substrates, the SUMOylation of target proteins can trigger extensive effects within cells [75,76]. The SUMOylation of specific proteins can affect the genome integrity, cell cycle, and transcription factor activity [77,78,79,80,81,82]. Moreover, SUMOylation can interactively affect the process of other PTMs like ubiquitination and acetylation [83,84,85].

Both ubiquitination and SUMOylation play important roles in various aspects of cellular or viral physiology. Within the following parts, we will summarize and discuss detailed regulation landscapes of ubiquitination- and SUMOylation-mediated retroviral infection and latency.

## 3. The Involvement of Ubiquitination and SUMOylation in Retroviruses Infection

### 3.1. Ubiquitination and Retroviruses Infection

As we have mentioned above, the ubiquitination procedure comprises three-step enzymatic reactions utilized by three enzymes, E1, E2, and E3, to covalently conjugate the Ub to protein substrates. The effect of the attachment of Ub to proteins depends on the types of linkage of the Ub chains. Retroviruses extensively exploit the host ubiquitination system to regulate viral replication and pathogenicity at each step of the viral life cycle (Figure 3, Table 1).

The life cycle of the retrovirus begins with the attachment of the viral envelope protein to specific receptors located on the surface of target cells. HIV-1 entry requires the binding of the viral envelope protein gp120 to the cellular CD4 receptor. Several viral proteins have been reported to downregulate the level of surface CD4 on HIV-1-infected cells via the Ub-mediated proteasomal degradation system [86]. The Vpu protein of HIV-1 interacts with the cytoplasmic tail of CD4, which contributes to the phosphorylation of amino acid di-serine motif (Ser52-Ser56) in Vpu, and then allows the recruitment of Beta-transducin repeats-containing protein (βTrCP), a subunit of the Skp1-cullin-F box (SCF) family of the ubiquitin ligase SCF^βTrCP^ [87,88]. Next, the SCF^βTrCP^ E3 ligase complex induces the K48-linked ubiquitination of the CD4 cytosolic tail on lysine residues, tagging CD4 for proteasomal degradation. Vpu is expressed during the late stage of viral infection and Vpu-mediated degradation of newly synthesized CD4 reduces the re-entry of newly released virions from infected cells and prevents super-infection of progeny virions. The evidence supports that the viral Nef is susceptible to the ubiquitination at Lys144, which is essential for Nef-mediated CD4 downregulation [89]. Additionally, the level of surface CD4 is downregulated by Nef-triggered endocytosis [90,91]. The chemokine receptors CCR5 and CXCR4, which function as co-receptors along with CD4 for the fusion of viral membranes with cellular membranes, can be down-modulated by viral-mediated proteasome pathway manipulation [92,93,94]. The possible reason for HIV-1 to downregulate the surface expression of its primary receptor/co-receptors is to avoid viral super-infection, prevent the premature cell death, and reduce the frequency of HIV-1 recombination. Taken together, all the evidence suggests that the number of cellular receptor/co-receptors on the surface of the infected cells, which is regulated by the viral proteins via ubiquitin-conjugating system during HIV-1 infection, is crucial for retrovirus entry as well as super-infection interference.

Due to the limited genomic capacity, retroviruses generally hijack the cellular machinery to complete the viral replication cycle and propagation. Meanwhile, the host cells have evolved versatile defense strategies to counteract retroviral replication utilizing a variety of host restrictive factors. Strikingly, most of the cellular factors restrict viral life cycles by targeting viral proteins for degradation by the ubiquitin-conjugating system. For instance, tripartite motif (TRIM) family proteins play extremely important roles in the innate immune defense, especially defending against the retroviral infection [95,96]. Tripartite motif protein 5 isoform alpha (TRIM5α) inhibits the post-entry infection of retroviruses in a species-specific manner by accelerating viral uncoating [97,98,99]. Once the virus enters into the cytoplasm, the retroviral capsid is captured by TRIM5α, which causes the capsid destabilization and susceptible to the proteasome or autophagy machinery, accelerates premature viral capsid disassembly, and thus negatively affects retroviral replication [98]. TRIM5α, as an E3 ubiquitin ligase, has been shown to undergo auto-ubiquitination, which increases its antiviral efficiency [100]. On the other hand, TRIM5α also functions as a pathogen recognition receptor and thus triggers antiviral innate immune response [101]. Particularly, the association of TRIM5α with susceptible retroviral capsids enhances its E3 ubiquitin ligase activity, which in turn recruits the E2-conjugating enzyme Ubiquitin-conjugating enzyme 13/Ubiquitin-conjugating Enzyme Variant 1A (UBC13/UEV1A), resulting in the K63-linked poly-ubiquitin chains and the auto-phosphorylation of cellular TAK1 [102]. Finally, TAK1 activates the downstream of AP-1- and NF-κB-signaling pathways, leading to IFN production and pro-inflammatory cytokines secretion against HIV-1 infection [102].

The Apolipoprotein B mRNA Editing Enzyme Catalytic Subunit (APOBEC) 3G is another retroviral restriction factor that blocks both nascent murine leukemia virus (MLV) and HIV-1 single-stranded cDNA synthesis during reverse transcription. Mechanistically, APOBEC3G is packaged into viral particles and mediates the lethal deamination of cytosine (C) to uracil (U) in nascent minus-strand viral cDNAs, leading to guanosine (G)-to-adenosine (A) hypermutation of the viral plus-strand DNA, resulting in the decreased viral infectivity [103]. In this respect, retroviruses have evolved a viral antagonist named Vif to counteract the host antiviral defense by hijacking the ubiquitin ligase complex [104,105]. It has been shown that Vif recruits the SCF-like complex, which consists of Cul5, elongin B, elongin C, and Rbx1, to induce the polyubiquitination and proteosomal degradation of cellular APOBEC3G, resulting in the abrogation of APOBEC3G-mediated restriction [106]. Intriguingly, the Vif-mediated degradation of APOBEC3G can be inhibited by both the cellular protein HSP70 and DUB USP49 [107,108]. Additionally, MLVs encode glycosylated Gag (glycol-Gag) protein to inhibit the access of APOBEC3 to the reverse transcription complex [109]. Other members of the APOBEC family, such as APOBEC3A, APOBEC3B, and APOBEC3F, also have been found to restrict HIV-1 and SIV replication, although they might not be as efficient as APOBEC3G [110,111,112]. The Vpx protein, the accessory protein of HIV-2, has been shown to counteract APOBEC3A, but the role of Vpx in counteracting APOBEC3A is still unclear [113]. Similarly, the SAM domain and HD domain-containing protein 1 (SAMHD1) inhibit viral infection by hydrolyzing the intracellular deoxynucleotide triphosphates (dNTP) to a lower level, which is insufficient for the synthesis of the viral genomic DNA during the viral reverse transcription [114]. However, the accessory proteins Vpr and Vpx recruit DCAF1, an adaptor of the CUL4A-DDB1-DCAF1 E3 ubiquitin ligase complex, which induces the polyubiquitination and proteasomal degradation of SAMHD1 [115,116,117]. Another study showed that the E3 ubiquitin ligase TRIM21 is also able to mediate the degradation of SAMHD1, resulting in the enhanced HIV-1 replication as well as IFN production [118]. Altogether, the host has utilized multiple restriction factors to inhibit the synthesis of viral genomic DNAs, while retroviruses also have evolved many accessory proteins to confront these restrictions by specifically inducing the degradation of antiviral factors.

The Integrase (IN) of retroviruses plays a crucial role in transporting the dsDNA of retroviruses into the nucleus and incorporating into the host chromosome. However, multiple lines of evidence suggest that the IN protein is susceptible to degradation by the host ubiquitin–proteasomal system (UPS). The N-terminal phenylalanine of IN is recognized by the UPS for rapid degradation, and the substitution of the N-terminal phenylalanine with the glycine, valine, or methionine increases the stability of IN [119,120]. Further investigation indicates that the cellular E3 RING ligase TRIM33 suppresses HIV-1 replication by promoting ubiquitin-/proteasome-mediated degradation of IN [121]. Conversely, both HIV-1 and MLV INs counteractively interact with the host protein Ku70, which specifically deubiquitinates IN and protects INs from polyubiquitination and proteasomal degradation [120,122].

Abundant evidence suggests that Ub is closely related to the retroviral budding and release [123,124,125]. Large amounts of free Ub and small amounts of mono-ubiquitylated Gag can be found within retroviral particles, including HIV-1, SIV, MLV, and avian leucosis virus (ALV) [125]. It has been found that proteasome inhibitors decreased the level of free Ub and prevented the mono-ubiquitination of Gag, which dramatically disrupted the release of retroviruses from infected cells [126]. Other studies suggested that virus release and proteolytic maturation require the mono-ubiquitination of Gag [127]. The p6 region of the retroviral Gag polyprotein harbors three distinct retroviral late domains, which include the Pro-Thr/Ser-Ala-Pro (PT/SAP) domain, the Pro-Pro-Pro-Tyr (PPPY) domain, and the Tyr-Pro-Xn-Leu (LYPXnL) domain. These domains play pivotal roles in the budding and release of progeny virions [128,129]. The PT/SAP domain within the HIV-1 Gag p6 region interacts with the putative ubiquitin regulator tumor susceptibility gene 101 (TSG101) [130], while the LYPXnL domain found in anemia virus Gag associates with ALG-2 interacting protein X (ALIX) [131]. Both TSG101 and ALIX belong to endosomal sorting complex required transport (ESCRT) machinery that is involved in remodeling membrane processes. These ESCRT factors are recruited by the late domains of Gag to proper membrane sites to assemble virions and mediate virus budding events. Moreover, the expression of the N-terminal of TSG101, which is also the Gag-binding domain, can inhibit the release of HIV-1 and MLV by disrupting the cellular endosomal sorting pathway [132,133]. The PPPY late domain is recognized by the E3 ubiquitin ligase NEDD4 that induces the ubiquitination of Gag, which facilitates the release of multiple retroviruses [134,135,136]. The presence of different late domains can increase the levels of Gag ubiquitination, which indicates the involvement of the host ubiquitination machinery in the late stages of the viral life cycle. Collectively, the current data show that different retroviral Gag proteins are widely ubiquitinated, which significantly influences the viral budding machinery.

The IFN-induced protein BST-2, which is also known as tetherin, exerts its antiviral effect on retroviral release by trapping the nascent virions on the cell plasma membrane after budding, preventing them from disseminating to other target cells [137,138]. Viruses have evolved specific strategies to displace tetherin from viral budding sites in order to facilitate viral release. For HIV-1, the accessory protein Vpu can directly interact with tetherin and reduce its distribution on the cellular surface, and thus promote viral release [138]. Like CD4 degradation, Vpu counters the tethering function of tetherin by interacting with β-TrCP to promote tetherin polyubiquitination and degradation in the endo-lysosomal system [139]. Further investigations show that Vpu also recruits the HRS protein, a key component of the ESCRT-0 complex, to contribute to the ubiquitination of tetherin, which further enhances the recruitment of tetherin by the ESCRT machinery and the subsequent degradation [140]. However, not all retroviruses encode Vpu. Some retroviruses have evolved other mechanisms to confront tetherin. SIV Nef proteins antagonize tetherin by hijacking the clathrin-mediated endocytosis, resulting in the downregulation of tetherin at the cell surface, which consequently facilitates virion release [141]. Moreover, HIV-2 Env and SIVtan Env are also found to counteract tetherin by reducing the distribution of cell surface tetherin [142,143]. It has been found that tetherin is also regulated by ubiquitination and lysosomal degradation mediated by E3 ubiquitin ligases MARCH8 and NEDD4, which provides better explanation for countering the suppression of tetherin on viral release [144].

### 3.2. SUMOylation and Retroviruses Infection

Like ubiquitination, the SUMOylation of target proteins is also involved in the regulation of retroviral infection. However, the biggest difference between ubiquitination and SUMOylation is that most ubiquitination on target proteins promotes their degradation, while SUMOylation on protein substrates represents, as signals, interactions with other proteins. Most SUMOylation-related retroviral studies focus on the life cycles of HIV-1, Moloney murine leukemia virus (MoMuLV), and HTLV. The SUMOylation pathway regulates retroviral infection by targeting both viral proteins and cellular factors, which will be summarized and discussed below (Figure 3, Table 1).

The HIV-1 p6 protein plays an important role in the formation of infectious viral particles, as well as the budding and release of progeny virions [145,146]. The p6 protein has been shown to undergo extensive PTMs, such as phosphorylation on Ser40, mono-ubiquitination on Lys27 and Lys33, and SUMOylation on Lys27, which significantly regulate its multifaceted function [127,147,148,149]. The p6 protein interacts with TSG101 and ALIX to assist the budding and scission of multiple vesicular bodies (MVBs) and promote membrane fission events [150]. SUMOylation and mono-ubiquitination of p6 share the same Lys27 residue. The mono-ubiquitination of p6 enhances the interaction between p6 and TSG101, as we have mentioned above, which promotes the budding of TSG101-related viruses [124]. The SUMOylation mediated by SUMO-1 and UBC9 at the same Lys27 residue of p6 protein prevents its ubiquitination to cripple the infection of virions [149]. However, HIV-1 bearing the p6-K27R mutation replicates like the wild-type HIV-1. Thus, how the ubiquitination and SUMOylation of p6 proteins influence the function of p6 needs to be further elucidated.

The retroviral IN protein mediates the integration of provirus into the host genome [151,152]. The IN proteins of both HIV-1 and the MoMuLV can be SUMOylated [153,154]. It has been found that IN can be SUMOylated on three Lys residues (K46, K136 and K244) [154]. Although the SUMOylation of IN does not influence its subcellular localization or stability, HIV-1 viral particles harboring SUMOylation site-mutated IN display reduced infectivity. Further mechanism studies reveal that the SUMOylation of IN does not affect its catalytic activity or the binding ability with its co-factor lens epithelium-derived growth factor p75 (LEDGF/p75). The reduced infectivity by mutating SUMOylation sites of IN mainly occurs before viral integration and after the reverse transcription. Remarkably, the SUMOylation of IN occurs in the next infection rather than during the viral production [155]. Apart from SUMOylation sites, the IN protein also harbors two SUMO-interacting motifs (SIMs), including SIM2 200IVDI203 and SIM3 257IKVV260 [156]. These SIMs are required for the interaction of IN with LEDGF/p75 but not with Ku70. Moreover, these two SIMs are also required for the nuclear localization of IN proteins. Mutation on these two sites significantly cripples HIV-1 infection among several steps of the life cycle, including the reverse transcription, the nuclear import, and the integration [156].

The presence of capsid (CA) in the PIC is important for early stages of retroviral infection [157]. The binding sites of SUMOylation E2 UBC9 and E3 PIASy have been found on CA proteins of MoMuLV, which facilitates the CA SUMOylation by SUMO-1 [158]. Further studies show that mutations of the binding sites, which interact with UBC9 and PIASy on CA, reduce or eliminate CA SUMOylation. The SUMOylation of MoMuLV CA is required for early events of viral infection, mainly before the nuclear entry of PIC and after the reverse transcription of viral RNA. Mutations of CA SUMOylation sites do not influence the late stage of viral gene expression or virion assembly [158].

The HTLV oncogenic protein Tax, which is a critical transcriptional activator of viral gene and able to initiate T-cell proliferation and promote tumorigenesis, can also undergo phosphorylation, ubiquitination, and SUMOylation [159,160]. Both HTLV-1 Tax-1 and HTLV-2 Tax-2 can be ubiquitinated and SUMOylated [161,162]. The SUMOylation of Tax is required for its binding to the Really Interesting New Gene Finger Protein 4 (RNF4), which ubiquitinylates Tax, resulting in the relocalization of Tax from the nucleus to the cytoplasm [163]. The SUMOylation of Tax was considered to be necessary for NF-κB activation in initial reports [160]. However, later studies showed that Tax SUMOylation was not a key determinant for Tax-mediated NF-κB activation. The non-SUMOylated Tax was still able to activate the NF-κB pathway [164]. Thus, the functional consequences of Tax SUMOylation are still uncertain.

The SUMOylation of cellular proteins also participates in the life cycle of retroviruses. The LEDGF/p75 is a transcriptional activator, which also acts as an HIV-1 integration co-factor through its chromatin binding capacity [165,166]. LEDGF/p75 interacts with the HIV-1 IN through its integrase binding domain (IBD) to facilitate integration events [167,168]. The SUMOylation of LEDGF/p75 in Lys364 residue mediated by UBC9 has no effect on the interaction of IBD with HIV-1 IN. However, the deSUMOylation of this residue attenuates the co-factor ability of LEDGF/p75 in proviral integration process, which indicates that LEDGF/p75 SUMOylation is essential for efficient retroviral integration [169].

The replication of HIV-1 can be regulated by the transcription factor NF-κB [170]. Normally, NF-κB localizes in the cytoplasm and is sequestered by the inhibitory protein IκBα. Upon the treatment of different stimuli, IκBα is phosphorylated and susceptible to poly-ubiquitination, which induces the degradation of IκBα and activates the release of NF-κB [171]. The ubiquitination and SUMOylation of IκBα share the same Lys21 residue on IκBα, which makes these two PTMs mutually incompatible. The SUMOylation of IκBα blocks its ubiquitination and enhances the suppression of NF-κB [172]. Therefore, the SUMOylation of IκBα can suppress the expression of NF-κB-related HIV-1 genes.

The nuclear factor of activated T cells (NFAT) is a transcriptional regulator for the LTR-mediated transcription, which is also modulated by the SUMOylation modification [81,173]. Physiologically, NFAT is retained in the cytoplasm in a resting state. The transcriptional activity of NFAT requires the SUMOylation at the Lys684 residue and Lys897 residue, which leads to the nuclear localization of NFAT [81]. In general, the histone deacetylases (HDACs) remove active marks acetyls from histone Lys residues, which contributes to the suppression of the HIV-1 transcription. The SUMOylation of Lys702 and Lys914 residues of NFAT facilitates the interaction of NFAT with HDACs, which promotes NFAT and HDACs to synergistically silence NFAT-targeted genes as well as HIV-1 LTR [174].

Most endogenous retroviruses (ERVs) are transcriptionally repressed by trimethylation at histone 3 lysine 9 (H3K9me3) and DNA methylation [175]. However, how the H3K9me3 modifiers, including SETDB1, SUV39H1, and SUV39H2, are recruited to the ERVs and maintain suppressive modifications is less defined. A systemic genome-siRNA screening work shows that SUMO2 and SUMO E3 ligase TRIM28 are key determinants of ERV silencing [176]. The auto-SUMOylated TRIM28 can bind SETDB1 and transcription co-factor hnRNP K to synergistically silence ERVs in a SUMOylation-dependent manner [177]. These studies indicate that the TRIM28-mediated SUMOylation is essential for H3K9me3 modifications of ERV LTR. In another protein network study, the SUMOylated GHKL ATPase protein Morc3 interacts with the histone H3.3 chaperone Daxx to silence ERV [178]. The SUMOylation-deficient Morc3 is unable to bind Daxx, which eventually reduces the histone H3.3 deposition on ERV. Future work needs to elucidate the relationship of Morc3 SUMOylation-mediated H3.3 incorporation and TRIM28 SUMOylation-mediated H3K9me3 establishment on the ERVs LTR.
viruses-15-00985-t001_Table 1Table 1Involvement of ubiquitination and SUMOylation of viral targets or cellular targets in retroviruses infection. The major function of PTMs of targets is also summarized. The order of these targets within the table is determined by the order that viral or cellular targets appear within the main text.PTMsVirus (s)Viral Target (s)Cellular Target (s)FunctionReferencesUbiquitinationHIV-1-CD4Inhibit viral entry and avoid super-infection[87]HIV-1Nef-Inhibit viral entry and avoid super-infection[89]HIV-1-TRIM5αAccelerate viral uncoating[98]HIV-1-TAK1Induce IFN production and pro-inflammatory cytokines secretion against HIV-1 infection[102]HIV-1, MLV-APOBEC3GPromote nascent viral single-stranded cDNA synthesis[106]HIV-1- SAMHD1Promote the synthesis of the viral genomic DNA[117]HIV-1, MLVIN-Suppress viral DNA integration and prevent provirus formation[121]HIV-1, SIV, MLV, ALVGag-Interfere the virion release of multiple retroviruses[130,136]HIV-1-TetherinPromote viral release[139,140]SUMOylationHIV-1p6-Decrease viral infectivity[149]HIV-1, MoMuLVIN-Ensure efficient infectivity[154]MoMuLVCA-Required for early events of viral infection and the formation of proviruses[158]HTLV-1Tax-Activate NF-κB pathway[160,164]HIV-1-LEDGF/p75Promote efficient viral integration[169]HIV-1-IκBαSuppress NF-κB-activated viral genes[172]HIV-1-NFATPromote nuclear localization of NFAT and silence NFAT-targeted genes[81,174]ERV-TRIM28Deposit SETDB1 and hnRNP K on ERV[177]ERV-Morc3Bind Daxx and promote H3.3 deposition on ERV[178]


## 4. The Involvement of Ubiquitination and SUMOylation in Retroviruses Latency

### 4.1. Ubiquitination and Retroviruses Latency

The integration of the retroviral genome into the host genome makes it possible to form a viral latent reservoir in which the viral genome is silenced and unrecognized by the host immune system. Similar to the role of ubiquitination on the retroviral life cycle, as described above, it is well-known that the host ubiquitination system is widely involved in retroviruses latency (Figure 4, Table 2). Low levels of HIV-1 Tat expression can lead to viral silencing and latency establishment, and recent studies show that HIV-1 Tat is strictly subjected to the regulation of UPS [179,180,181]. The Tat protein can hijack the E3 Ub ligases, such as proto-oncoprotein Hdm2, CHIP, and PJA2, as well as deubiquitinases USP21 and USP7 to regulate HIV-1 transcription by modulating Tat ubiquitination [182,183,184]. Besides the cellular proteins, the viral proteins, including Nef, Rev, and Nucleocapsid (NC) protein, also have been found to be involved in the regulation of ubiquitination and degradation of Tat. Nef interacts with Tat in the cytoplasm and promotes the degradation of Tat via the ubiquitin–proteasome-dependent pathway [185]. HIV-1 Rev triggers the degradation of Tat through reducing NAD(P)H: quinine oxidoreductase 1 (NQO1), which associates with and inhibits the 20S proteasome to degrade cellular proteins [180]. Moreover, NC can translocate to the nucleus and co-localize with Tat, which results in the degradation of Tat via UPS, thus inhibiting the transcription of HIV-1 [186]. Collectively, these findings have implicated the role of ubiquitination in controlling Tat-mediated transcription activation and eventually leading to the establishment of HIV-1 latency.

Our group has reported that the host factor ubiquitin-like PHD and RING finger domain 1 (UHRF1) maintains HIV-1 latency by mediating K48-linked ubiquitination and proteasomal degradation of Tat in a RING-dependent way [187]. Tat hijacks the positive transcriptional elongation factor b (p-TEFb), which consists of cyclin T1 and cyclin-dependent kinase 9 (CDK9), to mediate transcription elongation by phosphorylating the C-terminal domain of RNA Polymerase II (RNAP II), as well as the negative transcription factor NELF and DSIF [188,189,190]. UHRF1 interacts with HIV-1 Tat by competing with p-TEFb, leading to the disruption of the Tat/p-TEFb complex. Consequently, the HIV-1 transcription elongation is aborted [187]. In addition to p-TEFb, the cellular Super Elongation Complex (SEC), which consists of ELL2, AFF1/4, and ENL/AF9, is recruited by Tat to the promoter, leading to the release of paused RNAP II [191]. Like p-TEFb, ELL2 is essential for both basal and Tat-activated HIV-1 transcription. *Liu* et al. found that the E3 ubiquitin ligase Siah1 promotes the polyubiquitination and proteasomal degradation of ELL2. Intriguingly, the SEC scaffold protein AFF4 interacts with and sequesters ELL2 from Siah1, thereby preventing the polyubiquitination of ELL2. Further investigation showed that the latency reversal agents (LRAs) prostratin and HMBA significantly inhibit the expression of Siah1 and Siah1-mediated polyubiquitination of ELL2 [192].

Additionally, both lncRNAs and microRNAs have also engaged in modulating HIV-1 latency. Our previous study found that a cellular lncRNA NRON promotes Tat degradation through K48-linked Ub chain modification. NRON recruits the ubiquitin/proteasome components, including CUL4B and PSMD11. Thus, it potently maintains HIV-1 latency in resting CD4^+^ T cells [22]. The E3 ubiquitin ligase TRIM32 activates the NF-κB pathway by directly recruiting IκBα and ubiquitinating IκBα, thus resulting in the reactivation of HIV-1 latency. However, the miR-155 binds to TRIM32 mRNA and decreases the expression of TRIM32 to counter the reactivation of latent HIV-1 [193]. Altogether, these investigations highlight that ncRNAs and UPS are extensively involved in the regulation of HIV-1 latency.

In line with HIV-1 Tat, another transactivator named Tax encoded by HTLV-1 regulates the expression of proviral genes. Tax interacts with the CREB/ATF, the CBP/p300, and the Tax response element (TRE) located within the 5’ LTR promoter to enhance the transcription of proviral plus-strand DNA, thereby stimulating productive viral replication and virion production [194]. Recently, multiple lines of evidence support that Tax is subject to ubiquitination, which negatively regulates proviral gene expression. It is demonstrated that the E2 ubiquitin-conjugation enzyme UBC13 promotes the K63-linked ubiquitination of Tax and increases the interaction of Tax with NF-κB essential modulator (NEMO), one component of the IκB kinase complex, resulting in the NF-κΒ activation [195]. Furthermore, *Yan* et al. identified that the E3 ubiquitin ligase PDLIM2 can mediate the K48-linked ubiquitination of Tax and impose Tax to the nuclear matrix where the ubiquitinated Tax is degraded by the proteasome [196]. The HTLV-1 p13 protein is stabilized through the K48-dependent ubiquitination of serine and/or threonine residues, leading to the translocation of p13 to the nucleus and interfering with the Tax-CBP/p300 interaction, thereby inhibiting proviral transcription. These findings suggest that the ubiquitination of p13 may contribute to the establishment of the viral latent reservoir [197,198].

Several lines of evidence show that epigenetic modifications, such as histone acetylation, methylation, and DNA methylation, are involved in viral infection or the establishment of viral latency [199,200]. Recently, *Kulkarni* et al. found that the plus-strand transcription of HTLV-1 is governed by cellular p38-MAPK kinases, the glucose metabolism, and the ubiquitination of histone H2A (H2AK119ub1) [201]. Of note, H2AK119ub1 is enriched on the latent HTLV-1 promoter and provides an epigenetic barrier to the viral reactivation. Treatment with the general deubiquitinase inhibitor (DUBi) PR-619 results in the significant increase of H2AK119ub1 on the viral promoter and inhibits the proviral reactivation from latency [201]. These findings provide clues that epigenetic modifications including ubiquitination are involved in viral latency.

### 4.2. SUMOylation and Retroviruses Latency

The SUMOylation of viral or cellular proteins also play crucial roles in regulating retroviral latency (Figure 4, Table 2). The RNAP II is critical for the transcriptional initiation and elongation of the HIV-1 RNA [202]. For the transcriptional elongation, the p-TEFb catalytic subunit CDK9 phosphorylates the Ser2 residue of RNAP II, followed by releasing the stalled polymerase and intensely promoting the transcriptional elongation of HIV-1 [203]. To enter into the viral latency, CDK9- or RNAP II-mediated viral transcription must be terminated. The Lys44, Lys56, and Lys68 residues on CDK9 are susceptible to SUMOylation by the E3 ubiquitin ligase TRIM28, which results in the inhibition of CDK9 kinase activity and preventing p-TEFb assembly by blocking the interaction between CDK9 and Cyclin T1. Consequently, the HIV-1 latency is established [204]. Of note, we recently found that the histone chaperone CAF-1 forms nuclear bodies with liquid–liquid phase separation (LLPS) properties on the HIV-1 LTR, which recruits TRIM28 to SUMOylate CDK9 and maintains HIV-1 latency [205].

Most SUMOylated cellular proteins, such as the Polo-like kinase 1 (PLK1), the signal transducer and activator of transcription 5 (STAT5) and the breast cancer-associated gene 2 (BCA2), contribute to HIV-1 latency through their influence on HIV-1 transcription indirectly. PLK1 is critical for the regulation of the cell cycle and the signaling of cell survival [206,207,208]. Previous reports showed that the SUMOylation of PLK1 increases its nuclear translocation and protein stability [209]. Further study found that PLK1 is elevated by HIV-1 Nef upon HIV-1 reactivation, as well as *de novo* infection. More importantly, the infection of HIV-1 enhances PLK1 SUMOylation to stabilize PLK1 and promotes its nuclear localization, which inhibits viral cytopathic effects-induced cell death [210]. The living HIV-1-infected cells ultimately shape viral latent reservoirs. The STAT5 protein promotes HIV-1 replication through its phosphorylation, leading to the association of STAT5 with the HIV-1 LTR [211,212]. However, the SUMOylation of STAT5 inhibits its nuclear translocation and phosphorylation, which results in the inhibition of HIV-1 replication and promoting the establishment of HIV-1 latency [211]. Further investigation showed that benzotriazoles reactivate latent HIV-1 by inhibiting STAT5 SUMOylation, which indicates that benzotriazoles can be employed as promising LRAs [211]. BCA2 is an antiviral factor that can facilitate the degradation of HIV-1 particles via the tetherin-dependent pathway [213]. Another study showed that BCA2 also promotes Gag proteins degradation to inhibit virion production of HIV-1, SIV, and MoMuLV in a tetherin-independent way [214]. Additionally, BCA2 prevents the nuclear translocation of NF-κB by increasing the SUMOylation of IκBα, which enhances the hijacking activity of IκBα to NF-κB and eventually impairs the HIV-1 transcription [215]. Thus, BCA2 can also contribute to HIV-1 latency by inhibiting the NF–κB pathway.
viruses-15-00985-t002_Table 2Table 2Involvement of ubiquitination and SUMOylation of viral targets or cellular targets in retroviruses latency. The major function of PTMs of targets is also summarized. The order of these targets within the table is determined by the order that viral or cellular targets appear within the main text.PTMsVirus (s)Viral Target (s)Cellular Target (s)FunctionReferencesUbiquitinationHIV-1Tat-Disrupt HIV-1 transcription elongation[187]HIV-1-ELL2Disrupt HIV-1 transcription[192]HIV-1-BIRC2Activate NF–κB signaling and reactivate HIV-1 transcription[22]HIV-1-IκBαReactivate HIV-1 transcription[193]HTLV-1Tax-Activate NF–κB pathway by E2 UBC13-mediated K63-Ub[195]HTLV-1Tax-Degrade Tax by E3 PDLIM2-mediated K48-Ub[196]HTLV-1p13-Interfere Tax-CBP/p300 interaction and inhibit proviral transcription[197]HTLV-1-H2AInhibit provirus reactivation from latency[201]SUMOylationHIV-1-CDK9Reduce RNAP II activation to suppress HIV-1 transcription[204]HIV-1-TRIM28Maintain HIV-1 latency by coalescing with CAF-1[205]HIV-1-PLK1Prevent cell death of HIV-1-infected cells and increase the viral latent reservoir[208]HIV-1-STAT5Inhibit its nuclear translocation and promote HIV-1 latency[211,212]HIV-1-IκBαEnhance the hijacking activity of IκBα to NF–κB and impair HIV-1 transcription[214]HIV-1-PMLDegrade PML via ubiquitin-proteasome pathway in HIV-1 productively infected cells[216]HIV-1-SMC5/6Silence integration-competent HIV-1 proviruses[217]HIV-1-EZH2Mediate the formation of H3K27me3 and suppress HIV-1 transcription[218]


Some SUMOylated proteins influence HIV-1 latency by manipulating the epigenetic regulation. The promyelocytic leukemia protein nuclear bodies (PML NBs) play an important role in the maintenance of HIV-1 latency through interacting with the histone methyltransferase G9a [219]. Moreover, the functions of PML NBs on the chromatinization of both viral and cellular genomes highly depend on PML SUMOylation, as well as their LLPS properties [220,221]. However, another study revealed that the SUMOylation of PML NBs causes their degradation via the ubiquitin-proteasome pathway in HIV-1 productively infected cells [216]. Conversely, the interruption of the SUMOylation of PML NBs can stabilize the PML NBs and maintain the HIV-1 latency. Another protein complex, which can utilize the SUMOylation to epigenetically contribute to HIV-1 latency is the structural maintenance of chromosome (SMC) 5/6 complex. The host SMC 5/6 complex is involved in chromosomal replication, recombination, and repair [217]. The SUMO E3 ligase NSMCE2 is one of the components of the SMC 5/6 complex, which can SUMOylate several proteins as well as itself [217,222,223]. Some chromatin components, including the histone H4, are susceptible to SUMOylation. The SUMOylation of histones is associated with transcriptional repression [224,225]. Recently, *Irwan* et al. found the SMC5/6 complex plays an important role in mediating the SUMOylation of chromatinized unintegrated HIV-1 DNA, which contributes to the epigenetic silencing of unintegrated proviruses, and consequently promotes the establishment of HIV-1 latency [226]. Our group recently found that the Polycomb group (PcG) protein CBX4 forms nuclear bodies with LLPS properties and recruits EZH2 on the HIV-1 LTR [218]. CBX4 is also a SUMO E3 ligase, which can SUMOylate EZH2 with SUMO4. EZH2 mediates the histone lysine 27 (H3K27) methylation, resulting in the formation of suppressive mark H3K27me3 on the HIV-1 LTR [227]. The SUMOylation of EZH2 mediated by CBX4 strengthens the H3K27 methyltransferase activity of EZH2 [218], thereby potently contributes to HIV-1 latency. Taken together, the SUMOylation of cellular proteins can contribute to retroviral latency in both transcription-dependent and epigenetics-dependent ways.

## 5. Ubiquitination- and SUMOylation-Targeted Anti-Retroviral Drugs

The combination antiretroviral therapy (cART) has been successfully used to treat AIDS utilizing virus-targeted drugs to inhibit the process of HIV-1 replication, despite the evolvement of persistent drug resistance mutations during the treatment. The residual HIV-1 proviruses are temporarily silenced within CD4^+^ T cells from patients under cART and generally escape the immune recognition and the subsequent clearance, which forms the major obstacle to thoroughly eradicating HIV-1 [228]. Although the cART can achieve long-term remission of HIV-1 viremia, serious adverse reactions still exist, such as lactic acidosis, toxic hepatitis, hyperlipidemia, insulin resistance, and the accumulation of viral-resistance mutations or escape mutations [229]. To this end, the development of host-targeted drugs to purge latent reservoirs is a promising strategy for the treatment of retrovirus infection [230]. The latent retroviruses are, firstly, reactivated by specific molecules, which are called latency-reversing agents (LRAs). Then, the reactivated viruses or viral protein-expressing cells are eradicated by the host immune surveillance. The combination treatment of “reactivation” and “eradication” is also called the “shock and kill” strategy.

The Ubiquitination and the SUMOylation play crucial roles in regulating the infection and latency of retroviruses, as described above. Thus, the ubiquitination- and SUMOylation-targeted proteins may serve as potential targets for effective host-targeted anti-retroviral drugs. In available reports, the ubiquitination- and SUMOylation-targeted anti-retroviral drugs are rare. Benzotriazoles belong to a class of LRAs with the ability to reactivate latent HIV-1. In the presence of IL-2, benzotriazoles reactivate and reduce latent HIV-1 both in vitro and ex vivo by blocking the SUMOylation of STAT5 without the T cell activation or proliferation [211]. The PLK1 inhibitor SBE 13 HCl induces the cell death of HIV-1-infected cells by inhibiting both the kinase activity and the SUMOylation modification of PLK1, which indicates that PLK1 inhibitors could serve as efficient “kill” agents to eliminate reactivated HIV-1 and corresponding infected cells [210]. Except for inhibiting target modification, *Ikenna G. Madu* et al. have identified a chemotype of the small molecule inhibitor of SENP, which suppresses the HIV-1 integration through promoting the SUMOylation of IN [231]. These inhibitors do not inhibit viral production but inhibit viral infectivity by incorporating into the virions. Another study showed that the arsenic/interferon therapy can selectively induce apoptosis in adult T-cell leukemia/lymphoma (ATL) or the HTLV-1-transformed cells by driving the degradation of the HTLV-1 Tax. Under the arsenic/interferon treatment, Tax is recruited by PML NBs and SUMOylated by SUMO2/3, which results in the RNF4-mediated ubiquitination and proteasomal degradation of Tax [232]. Additionally, several lines of evidence show that compounds targeting the cellular ubiquitin ligases and deubiquitinases are attractive viral LRAs. A study found that several deubiquitinases, including UCH37, OTULIN, USP14, and USP5, contribute to HIV-1 latency. Of note, b-AP15, the dual inhibitor of UCH37 and USP14, reverses HIV-1 latency in a dose-dependent manner [233]. Furthermore, Debio 1143, the inhibitor of apoptosis protein antagonist (IAPa), induces the auto-ubiquitination and degradation of the ubiquitin ligase BIRC2, resulting in the activation of non-canonical NF–κB signaling, which eventually reactivates HIV-1 transcription [234]. Although the compounds mentioned above have only been extensively studied *in vitro*, abundant evidence shows that targeting the ubiquitination or SUMOylation process to regulate retroviral life cycle and latency has great potential for providing therapeutic options for retroviruses-related diseases in the future.

## 6. Conclusions

Significant advances have been made in elucidating the role of PTMs in retrovirus infection. In this review, we summarize the current progress of the mechanisms by which the ubiquitination and SUMOylation positively and negatively modulate the retroviral life cycle and latency, providing important insights into extensive interplays between retroviruses and cellular ubiquitination and SUMOylation systems. The presence of these interplays poses a major opportunity to target ubiquitination and SUMOylation systems as novel therapeutic strategies for retrovirus-related diseases. Of note, a SUMOylation inhibitor TAK-981 has shown potential effects in clinical trials for a broad range of cancers by targeting the SUMOylation pathway [235]. Intriguingly, the TAK-981 treatment also significantly promotes the transcription of unintegrated HIV-1 DNA and inhibits the establishment of HIV-1 latency in primary CD4^+^ T cells [226]. Collectively, these studies provide important insights for developing novel drugs based on the manipulation of ubiquitination and SUMOylation pathways to achieve a “sterilizing cure” or “functional cure” of retroviral infection.

## Figures and Tables

**Figure 1 viruses-15-00985-f001:**
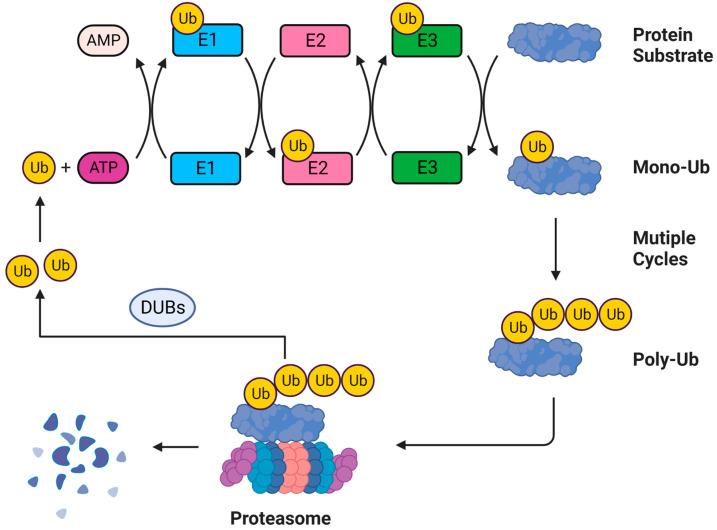
Schematic of the ubiquitination procedure. Firstly, the E1 enzyme utilizes the energy provided by ATP to activate the ubiquitin (Ub) to generate the Ub–E1 complex. The Ub from the Ub–E1 complex is transferred to the E2 enzyme to form the Ub–E2 complex. Subsequently, the E3 enzyme interacts with both the Ub–E2 complex and the protein substrate to conjugate Ub to the substrate. The substrate can undergo mono-ubiquitination or the poly-Ub chain reaction. The Ub on target proteins acts as signals to interact with other proteins or serves as degradation signals to degrade substrates with the proteasome system. Besides, the Ub on substrates can also be removed and reused by deubiquitylating enzymes (DUBs).

**Figure 2 viruses-15-00985-f002:**
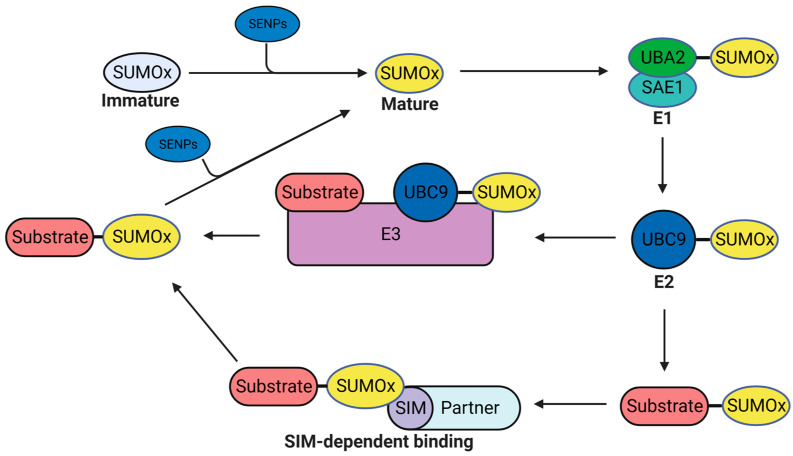
Schematic of the SUMOylation procedure. The immature precursor SUMO molecule is hydrolyzed by SUMO-specific proteases (SENPs) and processed into a mature form by presenting its C-terminal diglycine motif. Next, the mature form of SUMO is activated and transferred to the SUMO E2 enzyme UBC9 by the E1 enzyme complex (SAE1/UBA2). Then, UBC9 directly conjugates the SUMO molecule to the substrate protein or indirectly transfers the SUMO molecule to the substrate utilizing the SUMO E3 ligase. Most SUMO molecules on target proteins act as signals to interact with other partner proteins via the SUMO-interacting motif (SIM) presented on interacting proteins. The SUMO molecules can also be deconjugated from substrates by SENPs and reused by other protein substrates. SUMOx represents SUMO1, SUMO2, SUMO3, SUMO4, or SUMO5.

**Figure 3 viruses-15-00985-f003:**
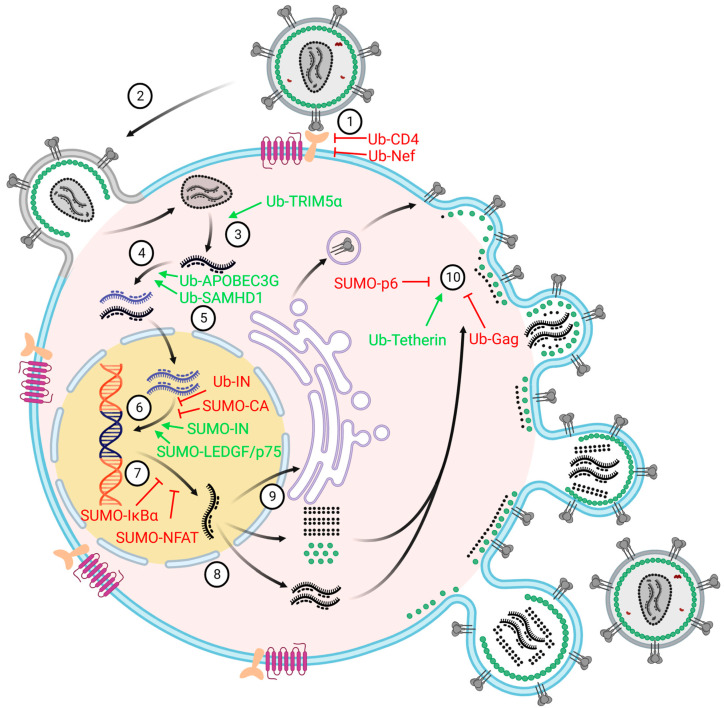
Schematic of the HIV-1 life cycle and corresponding restriction or activation factors. The numbered positions represent 10 steps of the HIV-1 life cycle, which include receptor binding, membrane fusion, viral uncoating, reverse transcription, nucleus importing, provirus integration, RNA transcription, RNA exporting, protein translation, and the virions formation. Each step is regulated by both ubiquitinated (Ub) proteins and SUMOylated (SUMO) proteins, which are shown beside the number. Proteins and arrows marked “red” represent inhibiting HIV-1 infection. Proteins and arrows marked “green” represent promoting HIV-1 infection. The exact functions of each protein can be found within the main text and Table 1.

**Figure 4 viruses-15-00985-f004:**
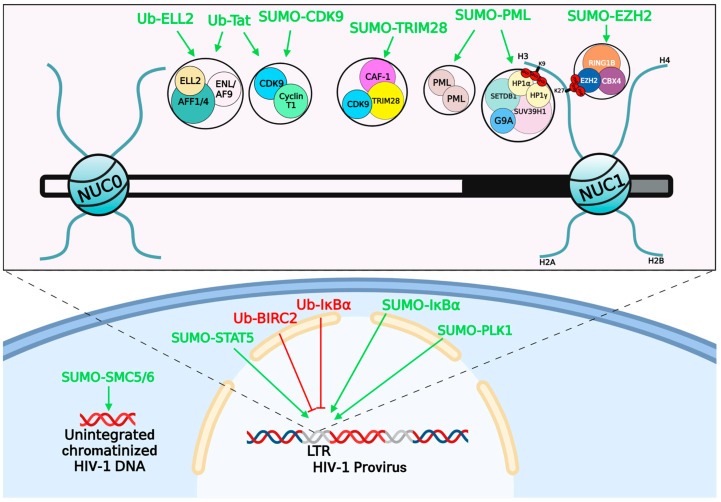
Schematic of HIV-1 latency and corresponding latency contributors and suppressors. The amplified region represents HIV-1 LTR where harbors the HIV-1 promoter. Protein complexes, which influence HIV-1 latency, are shown above the LTR. Proteins and arrows marked “red” represent inhibiting HIV-1 latency. Proteins and arrows marked “green” represent promoting HIV-1 latency. The exact functions of each protein can be found within the main text and Table 2.

## Data Availability

Not applicable.

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
