# Peer review of "The Involvement of Ubiquitination and SUMOylation in Retroviruses Infection and Latency"

_viruses, 2023, doi:10.3390/v15040985_

Round 1

Reviewer 1 Report

I think this is a very comprehensive listing of the literature related to ubiquitination and sumolyation in retroviral life cycle. However a review should provide an overview and allow readers to easily comprehend the literature available in this important topic. It should be also be informative for people not directly involved in study of retroviruses. In this regard it will be better if authors could summarize the literature/review using a schematics/flow charts to indicate the lifecycle stages of retroviruses and sequentially indicating highlighting the PTM’s regulation of those stages as inhibitory or activation. I think the review is too difficult to comprehend for a reader.

Author Response

Point-by-point response to each comment:

Reviewer 1:

Comments and Suggestions for Authors:

I think this is a very comprehensive listing of the literature related to ubiquitination and sumolyation in retroviral life cycle. However a review should provide an overview and allow readers to easily comprehend the literature available in this important topic. It should be also be informative for people not directly involved in study of retroviruses. In this regard it will be better if authors could summarize the literature/review using a schematics/flow charts to indicate the lifecycle stages of retroviruses and sequentially indicating highlighting the PTM’s regulation of those stages as inhibitory or activation. I think the review is too difficult to comprehend for a reader.

Reply: We appreciate for the reviewer’s suggestion and the support of our work. Accordingly, we replenish two schematic diagrams of the life cycle and the latency of HIV-1, which include multiple stages extensively modulated by ubiquitination and SUMOylation in Figure 3 and Figure 4 in the revised manuscript. Figure 3 represents the schematic of the HIV-1 life cycle and corresponding restriction or activation factors. Figure 4 represents the schematic of HIV-1 latency and corresponding latency contributors and suppressors.

Reviewer 2 Report

This paper summarizes the literature on the impact of ubiquinylation and SUMOylation on retrovirus replication, and on latency. Most of the known cases of these modifications, of either viral or host proteins, having an effect on any stage of replication, are presented. This will be a useful listing. It is perhaps disappointing that there is little to tie all these cases together – but perhaps there are no overarching generalities to be made. I see no glaring omissions.

Small points:
There is some need for editing for English grammar.

E.g. line 30: “While most ‘slow retroviruses’, such as the murine leukemia virus (MuLV), induce tumorigenesis within a long clinical latency [2]” – this is an incomplete sentence.

Line 32: This is an odd, inappropriate reference (#2, to the Abelson virus, an acute virus) for slow retroviruses.

And line 70: “The infection, expression and latency of retroviruses involve in multiple aspects” – grammar or meaning unclear.

And lines 72, 94, 96, 105, 365, 384, 420, 424: “While….” There is no subject/verb after the “While” clause .

Line 133: “E1, which acquires the ATP hydrolysis to supply the energy” – “acquires” is the wrong word.
Line 135: “which is also happened on the Cys residue of E2” – grammar.

Line 154: Whereas the function of the JAMM/MPN+ family relies on zinc ions.” Incomplete.

Line 183: “Despite some proteins can be SUMOylated”. Grammar.

Line 188: “Although the proportion of SUMOylated proteins is only less than 5% for a given substrate…” This is not always true. RANBP2 can be 50%. Many are below 1%.

Line 265: “resulting in the inability of retroviruses for effective reverse transcription” – vague. Be more specific.

Line 266: For completeness, it could be mentioned that MLVs use Glycogag to block APOBEC3.

Line 284: “TRIM21 is also able to mediate the degradation of SAMHD1 to inhibit HIV-1 expression” – too many double negatives. Degrading SAMHD1 would enhance HIV-1 RT.

Line 319: “The PPPY late domain is recognized by the E3 ubiquitin ligase NEDD4 that induces the ubiquitination of Gag and then interferes with the release of multiple retroviruses” – not right. The PPPY late domain is required for release (of MLVs, HTLV-1, and others).

Line 326: For tetherin, for completeness, it could be mentioned that SIVs use Nef to degrade tetherin. SIV agm and HIV-2 use their Env to do the same job.

Line 356: “Mono-ubiquitination of p6 is critical…” This is probably too strong. I think there are reports that ubiquitinylation of p6 is not required for replication (PMID 15613319 – “HIV-1 bearing the p6-K27R mutation replicated just like the wild type”.

Line 430: “TRIM28-coalesced” – wrong word.

Line 433: “sequentially” – wrong word
Table 1: SUMOylation of HIV-1 p6 is not so clearly said to reduce virus release. SUMOylation of MoMuLV CA is reported to be required for formation of provirus, not said to reduce it.

Author Response

Point-by-point response to each comment:

Reviewer 2:

Comments and Suggestions for Authors:

This paper summarizes the literature on the impact of ubiquinylation and SUMOylation on retrovirus replication, and on latency. Most of the known cases of these modifications, of either viral or host proteins, having an effect on any stage of replication, are presented. This will be a useful listing. It is perhaps disappointing that there is little to tie all these cases together – but perhaps there are no overarching generalities to be made. I see no glaring omissions.

Reply: We appreciate for reviewer’s comprehensive summaries and the support of our review.

Small points:

There is some need for editing for English grammar.

  1. E.g. line 30: “While most ‘slow retroviruses’, such as the murine leukemia virus (MuLV), induce tumorigenesis within a long clinical latency [2]” – this is an incomplete sentence.

Reply: We thank the reviewer for pointing out this inappropriate description. We have corrected this sentence by deleting the conjunction “While”. The revised new sentence is shown as below: “Most ‘slow retroviruses’, such as the murine leukemia virus (MuLV), induce tumorigene-sis within a long clinical latency [2].”

  1. Line 32: This is an odd, inappropriate reference (#2, to the Abelson virus, an acute virus) for slow retroviruses.

Reply: We sincerely apologize for the odd and inappropriate reference. We have cited the new relevant reference (PMID: 12665053) in our revised manuscript.

  1. And line 70: “The infection, expression and latency of retroviruses involve in multiple aspects” – grammar or meaning unclear.

Reply: We apologize for the grammar mistakes and unclear expression. We have modified it as below: “The infection, expression and latency of retroviruses involve multiple stages extensively modulated by the virus-host interplay.”

  1. And lines 72, 94, 96, 105, 365, 384, 420, 424: “While….” There is no subject/verb after the “While” clause.

Reply: We apologize for the unclear expression in our manuscript. We have revised the manuscript accordingly to ensure that the expression is suitable to be published. In most situations, the conjunction “while” is redundant. Thus, we have deleted this conjunction word. In other situations, we have replaced “While” with “However” to keep the sentence complete.

  1. Line 133: “E1, which acquires the ATP hydrolysis to supply the energy” – “acquires” is the wrong word.

Reply: We thank the reviewer for pointing out this inappropriate description. We have replaced it with “requires”.

  1. Line 135: “which is also happened on the Cys residue of E2” – grammar.

Reply: Thanks for pointing out this mistake. We have corrected this sentence as “The activated Ub is then transferred to the active site of the E2 enzyme, resulting in the conjugation of Ub on the Cys residue of E2” in our revised manuscript.

  1. Line 154: “Whereas the function of the JAMM/MPN+ family relies on zinc ions.” Incomplete.

Reply: We are sorry for this inappropriate description. We have revised it as “And the deubiquitylation function of the JAMM/MPN+ family relies on the coordination of zinc ions within the active sites”.

  1. Line 183: “Despite some proteins can be SUMOylated”. Grammar.

Reply: Thanks for pointing out this grammar mistake. We have revised it as “Although the SUMOylation modification can be found on many protein substrates” in our revised manuscript.

  1. Line 188: “Although the proportion of SUMOylated proteins is only less than 5% for a given substrate…” This is not always true. RANBP2 can be 50%. Many are below 1%.

Reply: We thank the reviewer for pointing out this inappropriate description. We have replaced it as “Although the proportion of SUMOylated proteins is only less than 5% for most protein substrates, the SUMOylation of target proteins can trigger extensive effects within cells”.

  1. Line 265: “resulting in the inability of retroviruses for effective reverse transcription” – vague. Be more specific.

Reply: Thanks for pointing out this inappropriate description. We corrected this sentence as below: “Mechanistically, APOBEC3G is packaged into viral particles and mediates the lethal de-amination of cytosine (C) to uracil (U) in nascent minus-strand viral cDNAs, leading to guanosine (G) to adenosine (A) hypermutation of the viral plus-strand DNA, resulting in the decreased viral infectivity”.

  1. Line 266: For completeness, it could be mentioned that MLVs use Glycogag to block APOBEC3.

Reply: We thank the reviewer for pointing out this constructive suggestion. We have added the sentence “Additionally, MLVs encode glycosylated Gag (glycol-Gag) protein to inhibit the access of APOBEC3 to the reverse transcription complex” and cite relevant reference in our revised manuscript.

  1. Line 284: “TRIM21 is also able to mediate the degradation of SAMHD1 to inhibit HIV-1 expression” – too many double negatives. Degrading SAMHD1 would enhance HIV-1 RT.

Reply: We apologize for this wrong description. We have carefully read the cited reference and found that we make the mistake, we have corrected “Another study showed that the E3 ubiquitin ligase TRIM21 is also able to mediate the degradation of SAMHD1, resulting in the enhanced HIV-1 expression as well as IFN production” in our revised manuscript.

  1. Line 319: “The PPPY late domain is recognized by the E3 ubiquitin ligase NEDD4 that induces the ubiquitination of Gag and then interferes with the release of multiple retroviruses” – not right. The PPPY late domain is required for release (of MLVs, HTLV-1, and others).

Reply: We thank the reviewer for pointing out this inappropriate description. We have revised the sentence as “The PPPY late domain is recognized by the E3 ubiquitin ligase NEDD4 that induces the ubiquitination of Gag, which facilitates the release of multiple retroviruses” in our revised manuscript.

  1. Line 326: For tetherin, for completeness, it could be mentioned that SIVs use Nef to degrade tetherin. SIV agm and HIV-2 use their Env to do the same job.

Reply: We appreciate the reviewer’s suggestion. Accordingly, we add the sentences “However, not all retroviruses encode Vpu. Some retroviruses have evolved other mechanisms to confront tetherin. SIVs Nef proteins antagonize tetherin by hijacking the clathrin-mediated endocytosis, resulting in the downregulation of tetherin at the cell surface, which consequently facilitates virion release [141]. Moreover, HIV-2 Env and SIVtan Env are also found to counteract tetherin by reducing the distribution of cell surface tetherin [142,143].” in our revised manuscript.

  1. Line 356: “Mono-ubiquitination of p6 is critical…” This is probably too strong. I think there are reports that ubiquitinylation of p6 is not required for replication (PMID 15613319– “HIV-1 bearing the p6-K27R mutation replicated just like the wild type”.

Reply: We thank the reviewer for pointing out this inappropriate description. We have revised these sentences as below: “The mono-ubiquitination of p6 enhances the interaction between p6 and TSG101 as we have mentioned above, which promotes the budding of TSG101-related viruses [124]. The SUMOylation mediated by SUMO-1 and UBC9 at the same Lys27 residue of p6 protein prevents its ubiquitination to cripple the infection of virions [149]. However, HIV-1 bearing the p6-K27R mutation replicates like the wild type HIV-1. Thus, how the ubiquitination and SUMOylation of p6 proteins influence the function of p6 needs to be further elucidated.”

  1. Line 430: “TRIM28-coalesced” – wrong word.

Reply: Thanks for pointing out this mistake. We have corrected as “the TRIM28-mediated SUMOylation” in our revised manuscript.

  1. Line 433: “sequentially” – wrong word

Reply: Thanks for pointing out this mistake. We have replaced “sequentially” as “eventually” in our revised manuscript.

  1. Table 1: SUMOylation of HIV-1 p6 is not so clearly said to reduce virus release. SUMOylation of MoMuLV CA is reported to be required for formation of provirus, not said to reduce it.

Reply: We thank the reviewer for pointing out these inappropriate descriptions. We have carefully read the references and corrected these sentences in our revised manuscript and table. For the SUMOylation of HIV-1 p6, we have described its function as: “Decrease viral infectivity”. For the SUMOylation of MoMuLV CA, we have described the function as: “Required for early events of viral infection and the formation of proviruses”.

Reviewer 3 Report

In the second paragraph of the introduction, a description of the classification of retroviruses is provided.  The information provided is incorrect.  Please review Retroviridae taxonomy for revision.  

In the third paragraph of the introduction, life cycle as described, is HIV and not applicable to other retroviruses, which have many very different mechanisms, ie. Reverse transcription prior to budding or prior to capsid release, and a variety of mechanisms for RNA export.  The authors should limit their review of retroviral life cycle to HIV or expand to include a much broader array of mechanisms.

The manuscript needs an English usage review.  A common example is the use of the article “the” before a process, such as “The Ubiquination”.  You can have “the ubiquitination of a specific protein”, but not “The ubiquitination is a process….”  Another example is “The ubiquitin”

A more uniform writing style throughout the document would be beneficial.

Author Response

Point-by-point response to each comment:

Reviewer 3:

Comments and Suggestions for Authors:

  1. In the second paragraph of the introduction, a description of the classification of retroviruses is provided.  The information provided is incorrect.  Please review Retroviridaetaxonomy for revision.

Reply: We apologize for this wrong information. We have carefully consulted the retroviridae taxonomy within Fields Virology and re-written the introduction of retroviridae as below: “Retroviruses were originally classified by the morphology of their virion cores which were visualized under the electron microscope. Based on the morphology, retroviruses were classified into four types including A-type, B-type, C-type, and D-type viruses. Recently, the retroviral genera have been formalized by the International Committee on Taxonomy of Viruses, which includes ‘simple’ retroviruses and ‘complex’ retroviruses. The simple viruses including alpharetroviruses, betaretroviruses and gammaretroviruses encode only the Gag, Pro, Pol and Env proteins. The complex viruses including deltaretroviruses, epsilonretroviruses, lentiviruses and spumaviruses also encode many regulatory proteins besides the above gene products.”

  1. In the third paragraph of the introduction, life cycle as described, is HIV and not applicable to other retroviruses, which have many very different mechanisms, ie. Reverse transcription prior to budding or prior to capsid release, and a variety of mechanisms for RNA export.  The authors should limit their review of retroviral life cycle to HIV or expand to include a much broader array of mechanisms.

Reply: We thank the reviewer for the kind suggestions. We also apologize for the inadequate descriptions. We have modified these sentences as below: “Different retroviruses have unique life cycles utilizing many different mechanisms. The reverse transcription procedure can happen prior to budding or prior to capsid release for different retroviruses. Different retroviruses also utilize a variety of mechanisms for RNA export. Since the life cycle of HIV-1 has been extensively investigated, we next introduce the life cycle of HIV-1 for illustration.”

  1. The manuscript needs an English usage review.  A common example is the use of the article “the” before a process, such as “The Ubiquination”.  You can have “the ubiquitination of a specific protein”, but not “The ubiquitination is a process….”  Another example is “The ubiquitin”

Reply: We apologize for the grammar mistakes. The language expression and any grammatical mistakes have been carefully checked and corrected. We revised the manuscript accordingly to ensure that the manuscript is suitable to be published.

  1. A more uniform writing style throughout the document would be beneficial.

Reply: Thanks for the reviewer’s kind suggestion. All the authors have carefully read and revised the manuscript. We have corrected many mistakes which we made in our original manuscript. In our revised manuscript, we checked carefully and modified all the writing style in order to be uniform to be published.